# Hierarchical Prompting Improves Visual Recognition On Accuracy, Data Efficiency and Explainability

## Abstract

When humans try to distinguish some inherently similar visual concepts, *e.g.*, Rosa Peace and China Rose, they may use the underlying hierarchical taxonomy to prompt the recognition. For example, given a prompt that the image belongs to the rose family, a person can narrow down the category range and thus focuses on the comparison between different roses. In this paper, we explore the hierarchical prompting for deep visual recognition (image classification, in particular) based on the prompting mechanism of the transformer. We show that the transformer can take the similar benefit by injecting the coarse-class prompts into the intermediate blocks. The resulting Transformer with Hierarchical Prompting (TransHP) is very simple and consists of three steps: 1) TransHP learns a set of prompt tokens to represent the coarse classes, 2) learns to predict the coarse class of the input image using an intermediate block, and 3) absorbs the prompt token of the predicted coarse class into the feature tokens. Consequently, the injected coarse-class prompt conditions (influences) the subsequent feature extraction and encourages better focus on the relatively subtle differences among the descendant classes. Through extensive experiments on popular image classification datasets, we show that this simple hierarchical prompting improves visual recognition on classification accuracy (*e.g.*, improving ViT-B/16 by $+2.83\%$ ImageNet classification accuracy), training data efficiency (*e.g.*, $+12.69\%$ improvement over the baseline under $10\%$ ImageNet training data), and model explainability.

## 1 Introduction

For human visual recognition, awareness of the underlying semantic hierarchy is sometimes beneficial, especially when the object is difficult to recognize. More specifically, when trying to distinguish some inherently similar visual concepts, a person may use the hierarchical taxonomy to prompt the recognition. For example, the China Rose is easily confused with the Rosa Peace when the scope-of-interest is the whole Plantae (or even larger). However, given the prompt that the image belongs to the rose family (*i.e.*, the ancestor class), a person can narrow down the category range and shift his/her focus to the subtle variation between different roses. Therefore, the prompt of the coarse (ancestor) class in the hierarchy conditions (influences) the subsequent inference and benefits the fine (descendant) class recognition.

In this paper, we explore the above hierarchical prompting for deep visual recognition. We base our exploration on the prompting mechanism of the transformer, which typically uses prompt to condition the model for different tasks (Li & Liang, 2021; Gu et al., 2021; He et al., 2021), different domains (Ge et al., 2022), *etc*. For the first time, we show that in the image classification task, the transformer can benefit from being prompted with coarse class information. To this end, we inject the coarse-class prompts into the intermediate block to dynamically condition the subsequent feature extraction. Such a hierarchical prompting is similar as in the human visual recognition.

Specifically, exploiting the underlying semantic hierarchy to improve visual recognition has attracted great research interest and yielded several popular tasks, e.g., hierarchical image classification and hierarchical semantic segmentation. Considering that classification is fundamental for many computer vision tasks, this paper focuses on hierarchical image classification. Many popular image classification datasets (e.g., ImageNet and iNaturalist) can well accommodate this task because they already provide hierarchical annotations ("coarse + fine" labels). Compared with prior literature on

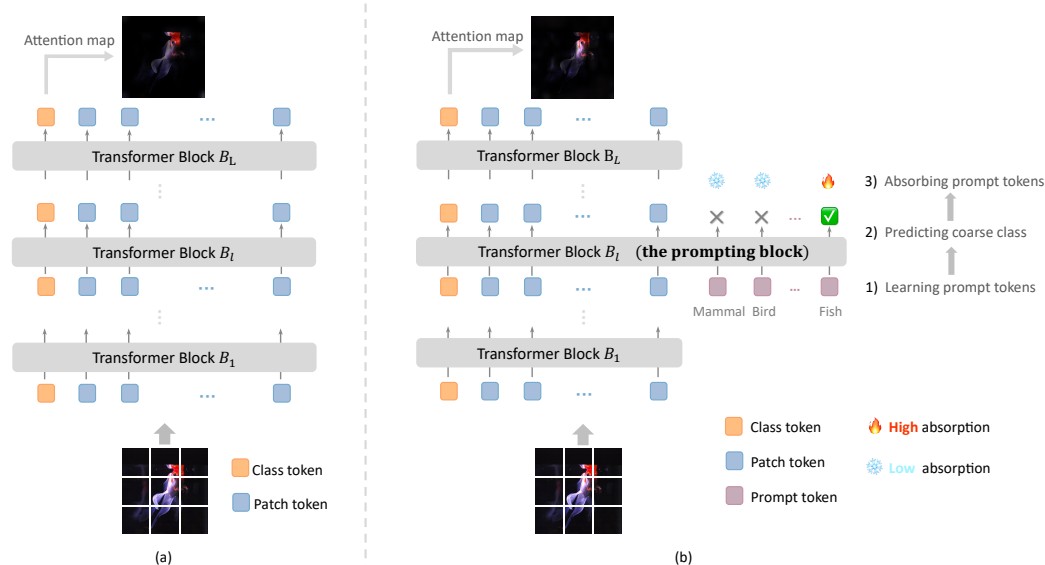

Figure 1: The comparison between Vision Transformer (ViT) and the proposed Transformer with Hierarchical Prompting (TransHP). In (a), ViT attends to the overall foreground region and recognizes the goldfish from the 1000 classes in ImageNet. In (b), TransHP uses an intermediate block to recognize the input image as belonging to the fish family and then injects the corresponding prompt. Afterwards, the last block attends to the face and crown which are particularly informative for distinguishing the goldfish against other fish species. Please refer to Fig. 5 for more visualizations. Note that TransHP may have several prompting blocks, and we only add one in (b) for demonstration.

this topic, our method has significant differences due to the employed prompting mechanism. Please refer to Section 2 (Related Works) for a detailed comparison.

We model our intuition into a Transformer with Hierarchical Prompting (TransHP) based on Vision Transformer (ViT) (Fig. 1 (a)). TransHP is very simple, as illustrated in Fig. 1 (b). Without loss of generality, Fig. 1 assumes the hierarchy has only two levels for simplicity, *i.e.*, a coarse level and a fine level. In other word, each image has a coarse label (*e.g.*, the fish) and a fine label (*e.g.*, the goldfish), simultaneously. TransHP selects an intermediate block as the "prompting block" to inject the coarse class information. Specifically, given the feature tokens (*i.e.*, the "class" token and the patch tokens) output from the preceding block, the prompting block concatenates them with a set of prompt tokens. Each prompt token represents a coarse class and is learnable (Section 3.2). The prompting block learns to predict the coarse class of the input image and to select the corresponding prompt token through weighted absorption (*i.e.*, high absorption on the target prompt and low absorption on the non-target prompt). Therefore, during inference, the prompt injection concentrates on the predicted coarse class on the fly and dynamically conditions the subsequent recognition.

Since our prompting mechanism follows the coarse-to-fine (or ancestor-to-descendant) semantic structure, we term it as the hierarchical prompting. We hypothesize this hierarchical prompting (and conditioning) will encourage TransHP to focus on the subtle differences among the descendant classes for better discrimination. Fig. 1 partially validates our hypothesis by visualizing the attention map of the class token in the last transformer block. In Fig. 1 (a), given a goldfish as the input image, the baseline model (ViT) attends to the whole body for recognizing it from the entire 1000 classes in ImageNet. In contrast, in TransHP in Fig. 1 (b), since the intermediate block has already received the prompt of "fish", the final block mainly attends to the face and crown which are particularly informative for distinguishing the goldfish against other fish species. Please refer to Section 4.4 for more visualization examples.

We conduct extensive experiments on multiple image classification datasets (*e.g.*, ImageNet (Deng et al., 2009) and iNaturalist (Van Horn et al., 2018)) and show that the hierarchical prompting improves the accuracy, data efficiency and explainability of the transformer: **(1) Accuracy.** TransHP

brings consistent improvement on multiple popular transformer backbones and five image classification datasets. For example, on ImageNet, TransHP improves ViT-B/16 (Dosovitskiy et al., 2021) by +2.83% top-1 accuracy. **(2) Data efficiency.** While reducing the training data inevitably compromises the accuracy, TransHP maintains better resistance against the insufficient data problem. For example, when we reduce the training data of ImageNet to 10%, TransHP enlarges its improvement over the baseline to +12.69%. **(3) Explainability.** Through visualization, we observe that the proposed TransHP shares some similar patterns with human visual recognition (Johannesson et al., 2016; Mirza et al., 2018), *e.g.*, taking an overview for coarse recognition and then focusing on some critical local regions for the subsequent recognition after prompting.

## 2    RELATED WORKS

**Hierarchical visual recognition** is based on the hierarchy underlying the visual concepts. Many works have explored hierarchical labels for improving the final fine-level classification. For example, Guided (Landrieu & Garnot, 2021) integrates a cost-matrix-defined metric into the supervision of a prototypical network. HiMulConE (Zhang et al., 2022a) builds an embedding space in which the distance between two classes are roughly consistent with the hierarchy (*e.g.*, two sibling classes sharing a same ancestor are relatively close and the classes with different ancestors are far away).

This paper has fundamental differences from these prior works from the viewpoint of the learned mapping function. Specifically, a deep visual recognition model can be viewed as a mapping function from the raw image space into the label space. All these prior methods learns a shared mapping for all the images to be recognized. In contrast, the proposed TransHP uses the coarse-class prompt to condition itself (from an intermediate block). It can be viewed as specifying an individual mapping for different coarse classes, yielding a set of mapping functions. Importantly, TransHP makes all these mapping functions share a same transformer, and conditions the single transformer into different mapping functions through the prompting mechanism.

**Prompting** was first proposed in NLP tasks (Brown et al., 2020; Gao et al., 2021; Jiang et al., 2020), and then has drawn research interest from the computer vision community, e.g. continual learning (Wang et al., 2022b;a), image segmentation (Lüddecke & Ecker, 2022), and neural architecture search (Zhang et al., 2022b). VPT (Jia et al., 2022) focuses on how to fine-tune pre-trained ViT models to downstream tasks efficiently. Prompting can efficiently adapt transformers to different tasks or domains while keeping the transforms' parameters untouched.

Based on the prompting mechanism, our hierarchical prompting makes some novel explorations, *w.r.t.* the prompting objective, prompting structure, prompt selection manner, and training process. 1) Objective: previous methods usually prompt for different tasks or different domains. In contrast, TransHP prompts for coarse classes in the hierarchy, in analogy to the hierarchical prompting in human visual recognition. 2) Structure: previous methods usually inject prompt tokens to condition the whole model. In contrast, in TransHP, the bottom blocks is completely shared, and the prompt tokens are injected into the intermediate blocks to condition the subsequent inference. Therefore, the prompting follows a hierarchical structure in accordance to the semantic hierarchy under consideration. 3) Prompt selection. TransHP pre-pends all the prompt tokens for different coarse classes and autonomously selects the prompt of interest, which is also new (as to be detailed in Section 3.3). 4) Training process. The prompting technique usually consists of two stages, *i.e.*, pre-training a base model and then learning the prompts for novel downstream tasks. When learning the prompt, the pre-trained model is usually frozen. This pipeline is different from our end-to-end pipeline, *i.e.* no more fine-tuning after this training.

## 3    TRANSFORMER WITH HIERARCHICAL PROMPTING

We first revisit a basic transformer for visual recognition (ViT (Dosovitskiy et al., 2021)) and the general prompting technique in Section 3.1. Afterwards, we illustrate how to reshape an intermediate block of the backbone into a hierarchical prompting block for TransHP in Section 3.2. Finally, we investigate how the prompting layer absorbs the prompt tokens into the feature tokens in Section 3.3.

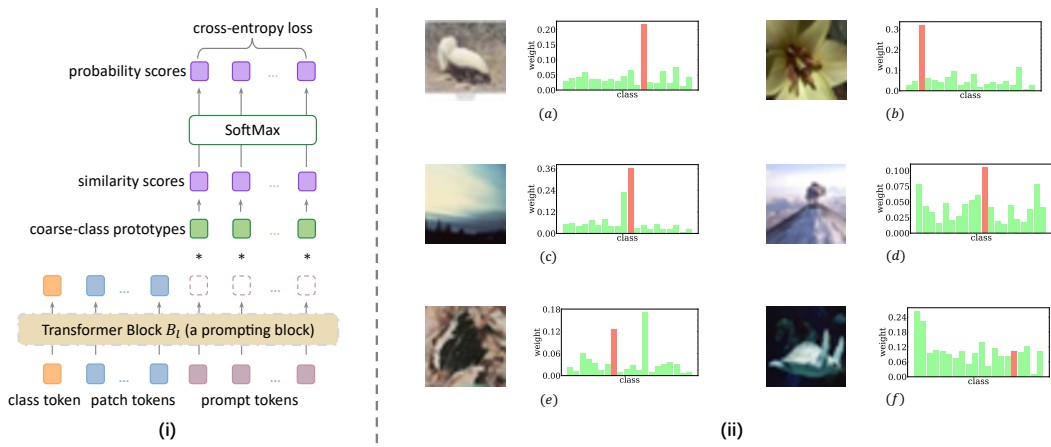

Figure 2: **(i)** illustrates the structure of a prompting block in TransHP. Instead of manually selecting the prompt of the coarse class, our prompting block pre-pends the whole prompt pool consisting of $M$ prompts ($M$ is the number of coarse classes) and performs autonomous selection. Specifically, it learns to predict the coarse class (Section 3.2) and spontaneously selects the corresponding prompt for absorption through soft weighting (Section 3.3), *i.e.*, the predicted class has the largest absorption weight. **(ii)** visualizes the absorption weights of all the 20 coarse-class prompts for some CIFAR-100 images. It shows how TransHP selects the prompts when the coarse class prediction is correct ($a$ and $b$), ambiguous ($c$ and $d$), and incorrect ($e$ and $f$), respectively. The red and green columns correspond to the true and false classes, respectively. The detailed investigation is in Section 3.3.

## 3.1 Preliminaries for Vision Transformer and Prompting

**Vision Transformer** (ViT) first splits an image into $N$ patches ($\left\{x_i \in \mathbb{R}^{3 \times P \times P} \mid i = 1, 2, \ldots, N\right\}$, where $P \times P$ is the patch size) and then embeds each patch into a $C$-dimensional embedding by $\mathbf{x}_i = \text{Embed}(x_i)$. Afterwards, ViT concatenates a class token $x^0_{cls} \in \mathbb{R}^C$ to the patch tokens and feed them into the stacked transformer blocks, which is formulated as:

$$\left[\mathbf{x}^l_{cls}, \mathbf{X}^l\right] = B_l \left(\left[\mathbf{x}^{l-1}_{cls}, \mathbf{X}^{l-1}\right]\right), \quad l = 1, 2, \ldots, L \tag{1}$$

where $\mathbf{x}^l_{cls}$ and $\mathbf{X}^l$ are the class token and the patch tokens after the $l$-th transformer block $B_l$, respectively. After the total $L$ blocks, the final state of the class token ($\mathbf{x}^L_{cls}$) is viewed as the deep representation of the input image and is used for class prediction. In this paper, we call the concatenation of class token and patch tokens (*i.e.*, $\left[\mathbf{x}^{l-1}_{cls}, \mathbf{X}^{l-1}\right]$) as the feature tokens.

**Prompting** was first introduced in Natural Language Processing to switch the same transformer model for different tasks by inserting a few hint words into the input sentences. More generally, it conditions the transformer to different tasks, different domains, *etc*, without changing the transformer parameters but only changing the prompts. To condition the model for the $k$-th task (or domain), a popular practice is to select a prompt $\mathbf{p}_k$ from a prompt pool $\mathbf{P} = \{\mathbf{p}_0, \mathbf{p}_1, \cdots\}$ and pre-pend it to the first block. Correspondingly, Eqn. 1 turns into:

$$\left[\mathbf{x}^l_{cls}, \mathbf{X}^l, \mathbf{p}^l_k\right] = B_l \left(\left[\mathbf{x}^{l-1}_{cls}, \mathbf{X}^{l-1}, \mathbf{p}^{l-1}_k\right]\right), \tag{2}$$

where $\mathbf{p}_k \in \mathbf{P}$ (the superscript is omitted) conditions the transformer for the $k$-th task.

## 3.2 The Prompting Block of TransHP

The proposed TransHP selects an intermediate transformer block $B_l$ and reshapes it into a prompting block for injecting the coarse-class information. Let us assume that there are $M$ coarse classes. Correspondingly, TransHP uses $M$ learnable prompt tokens $\mathbf{P}_M = [\mathbf{p}_0, \mathbf{p}_1, ..., \mathbf{p}_{M-1}]$ to represent these coarse classes. Our intention is to inject $\mathbf{p}_k$ into the prompting layer, if the input image belongs to the $k$-th coarse class.

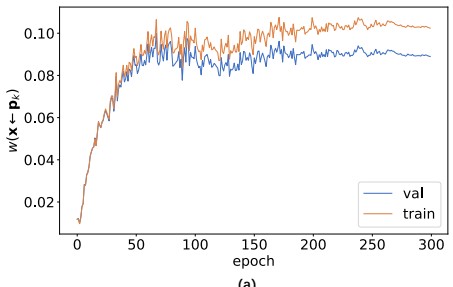 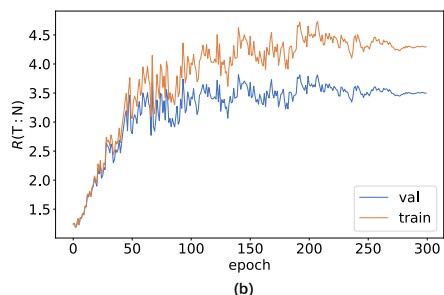

(a)  (b)

Figure 3: TransHP gradually focuses on the predicted coarse class when absorbing the prompts, yielding an autonomous selection. (a) The absorption weight of the target prompt. (b) The ratio of the target prompt weight against the largest non-target prompt weight (Eqn. 8). The dataset is CIFAR-100. We visualize these statistics on both the training and validation sets.

Instead of manually selecting the $k$-th prompt $\mathbf{p}_k$ (as in Eqn. 2), TransHP pre-pends the whole prompting pool $\mathbf{P}_M = [\mathbf{p}_0, \mathbf{p}_1, ..., \mathbf{p}_{M-1}]$ to the prompting layer and makes the prompting layer automatically select $\mathbf{p}_k$ for absorption. Specifically, through our design, TransHP learns to automatically 1) predict the coarse class, 2) select the corresponding prompt for absorption through "soft weighting", *i.e.*, high absorption on the target prompt and low absorption on the non-target prompts. The learning procedure is illustrated in Fig. 2 (i). The output of the prompting layer is derived by:

$$\left[\mathbf{x}_{cls}^l, \mathbf{X}^l, \hat{\mathbf{P}}_M\right] = B_l\left(\left[\mathbf{x}_{cls}^{l-1}, \mathbf{X}^{l-1}, \mathbf{P}_M\right]\right), \tag{3}$$

where $\hat{\mathbf{P}}_M$ is the output state of the prompt pool $\mathbf{P}_M$ through the $l$-th transformer block $B_l$. $\hat{\mathbf{P}}_M$ will not be further forwarded into the following block. Instead, we use $\hat{\mathbf{P}}_M$ to predict the coarse classes of the input image. To this end, we compare $\hat{\mathbf{P}}_M$ against a set of coarse-class prototypes and derive the corresponding similarity scores by:

$$S = \{\mathbf{p}_i^{\mathrm{T}}\mathbf{w}_i\}, i = 1, 2, \cdots, M, \tag{4}$$

where $\mathbf{w}_i$ is the learnable prototype of the $i$-th coarse class. We further use a softmax plus cross-entropy loss to supervise the similarity scores, which is formulated as:

$$\mathcal{L}_{\mathrm{coarse}} = -log\frac{\mathbf{p}_y^{\mathrm{T}}\mathbf{w}_y}{\sum_{i=1}^{M}\exp\left(\mathbf{p}_i^{\mathrm{T}}\mathbf{w}_i\right)}, \tag{5}$$

where $y$ is the coarse label of the input image. We note there is a difference between the above coarse classification and the popular classification: the popular classification usually compares a single representation against a set of prototypes. In contrast, our coarse classification conducts a set-to-set comparison (*i.e.*, $M$ tokens against $M$ prototypes). Note that TransHP may have several prompting blocks. We only introduce one prompting block in this section, and other blocks follow a similar procedure. The whole picture is demonstrated in Appendix A.1.

Through the above training, the prompting layer explicitly learns the coarse-class prompts, as well as predicting the coarse class of the input image. Given all the prompts $\mathbf{P}_M = \{\mathbf{p}_1, \mathbf{p}_2, \cdots, \mathbf{p}_M\}$ and the predicted coarse class $k = \underset{i}{\mathrm{argmax}}\left(\mathbf{p}_i^{\mathrm{T}}\mathbf{w}_i\right)$, TransHP will spontaneously select $\mathbf{p}_k$ for prompt absorption in a soft weighting manner, as to be explained in the following Section 3.3.

### 3.3  INVESTIGATING THE PROMPT ABSORPTION

The self-attention mechanism in the transformer allows each token to absorb information from all the tokens (including the feature tokens and the prompt tokens). In this section, we investigate how much the target prompt (*i.e.*, the prompt of the predicted coarse class) is absorbed through self-attention. In Eqn. 3, given a feature token $\mathbf{x} \in [\mathbf{x}_{cls}, \mathbf{X}]$ (the superscript is omitted for simplicity), we derive its absorption weights from the self-attention, which is formulated as:

$$W(\mathbf{x}) = \mathrm{Softmax}\left(\frac{Q(\mathbf{x})^{\mathrm{T}} K([\mathbf{x}_{cls}, \mathbf{X}, \mathbf{P}_M])}{\sqrt{d_k}}\right), \tag{6}$$

where $Q()$ and $K()$ are the operation that maps the input tokens into query and keys, respectively. $d_k$ is the scale factor. More specifically, in $W(\mathbf{x})$, the absorption weight on the $i$-th prompt token is calculated by:

$$w(\mathbf{x} \leftarrow \mathbf{p}_i) = \frac{\exp\left(Q(\mathbf{x})^{\mathrm{T}} K(\mathbf{p}_i)/\sqrt{d_k}\right)}{\sum \exp\left(Q(\mathbf{x})^{\top} K([\mathbf{x}_{cls}, \mathbf{X}, \mathbf{P}_M])/\sqrt{d_k}\right)}. \tag{7}$$

Based on the absorption weights, we consider two statistics:

- The absorption weight of the target prompt, *i.e.*, $w(\mathbf{x} \leftarrow \mathbf{p}_k)$.
- The ratio of the target prompt score to the largest non-target prompt score:

$$R(\mathtt{T:N}) = w(\mathbf{x} \leftarrow \mathbf{p}_k)/\max\{w(\mathbf{x} \leftarrow \mathbf{p}_{i \neq k})\}. \tag{8}$$

The target prompt weight $w(\mathbf{x} \leftarrow \mathbf{p}_k)$ indicates the importance of the target prompt among all the tokens. $R(\mathtt{T:N})$ measures the importance of the target prompt compared with the most prominent non-target prompt. Fig. 3 visualizes these statistics at each training epoch on CIFAR-100 (Krizhevsky et al., 2009), from which we make two observations:

**Remark 1: The importance of the target prompt gradually increases to a high level.** From Fig. 3 (a), it is observed that the absorption weight on the prompt token undergoes a rapid increase and finally reaches about 0.09. We note that 0.09 is significantly larger than the averaged weight 1/217 (there are 1 class token + 196 patch tokens + 20 prompt tokens).

**Remark 2: The target prompt gradually dominates among all the prompts.** From Fig. 3 (b), it is observed that although the absorption weight on the target prompt is close to the non-target prompt weight at the start, it gradually becomes much larger than the non-target prompt weight (about $4\times$).

Combining the above two observations, we infer that during training, the prompting block of TransHP learns to focus on the target prompt $\mathbf{p}_k$ (within the entire prompt pool $\mathbf{P}_M$) for prompt absorption (Remark 2), yielding a soft-weighted selection on the target prompt. This dynamic absorption on the target prompt largely impacts the self-attention in the prompting layer (Remark 1) and conditions the subsequent feature extraction.

We further visualize some instances for intuitively understanding the prompt absorption in Fig.2 (ii). Specifically, we visualize the absorption weights on all the 20 prompts for CIFAR-100. We note that the coarse prediction may sometimes be incorrect. Therefore, we use the red (green) column to mark the prompts of the true (false) coarse class, respectively. In (a) and (b), TransHP correctly recognizes the coarse class of the input images and makes accurate prompt selection. The prompt of the true class has the largest absorption weight and thus dominates the prompt absorption. In (c) and (d), TransHP encounters some confusion for distinguishing two similar coarse classes (due to their inherent similarity or image blur), and thus makes ambiguous selection. In (e) and (f), TransHP makes incorrect coarse-class prediction and correspondingly selects the prompt of a false class as the target prompt. The names of 20 coarse classes are in Appendix A.2.

## 4 EXPERIMENTS

### 4.1 IMPLEMENTATION DETAILS

Please refer to Appendix A.3.

### 4.2 TRANSHP IMPROVES THE ACCURACY

**Improvement on ImageNet and the ablation study.** We validate the effectiveness of TransHP on ImageNet and conduct the ablation study by comparing TransHP against two variants, as well as the baseline. As illustrated in Fig. 4, the two variants are: 1) we do not inject any prompts, but use

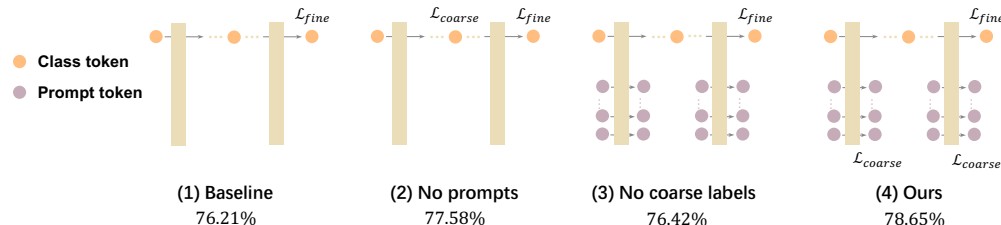

Figure 4: Comparison between TransHP and its variants (and the baseline) on ImageNet. 1) A variant uses the coarse labels to supervise the class token in the intermediate layers (No prompts). 2) A variant injects additional tokens without supervision from the coarse-class labels (No coarse labels). 3) TransHP injects coarse-class information through prompt tokens and achieves the largest improvement (Ours).

Table 1: The top-1 accuracy of TransHP on some other datasets (besides ImageNet). "w Pre" or "w/o Pre" denotes the models are trained from ImageNet pre-training or from scratch, respectively.

| Accuracy (%) | iNaturalist-2018 | iNaturalist-2019 | CIFAR-100 | DeepFashion |
|---|---|---|---|---|
| Baseline (w/o Pre) | 51.07 | 57.33 | 61.77 | 83.42 |
| TransHP (w/o Pre) | 53.22 | 59.24 | 67.09 | 85.72 |
| Baseline (w Pre) | 63.01 | 69.31 | 84.98 | 88.54 |
| TransHP (w Pre) | 64.21 | 71.62 | 86.85 | 89.93 |

the coarse labels to supervise the class token in the intermediate layers: similar with the final fine-level classification, the class token is also used for coarse-level classification. 2) we inject learnable tokens, but do not use the coarse labels as their supervision signal. Therefore, these tokens do not contain any coarse class information. From Fig. 4, we draw three observations as below:

**1)** Comparing TransHP against the baseline, we observe a clear improvement of $+2.44\%$ top-1 accuracy, confirming the effectiveness of TransHP on ImageNet classification. **2)** Variant 1 ("No prompts") achieves some improvement ($+1.37\%$) over the baseline as well, but is still lower than TransHP by $-1.07\%$. It shows that using the hierarchical labels to supervise the intermediate state of the class token is also beneficial. However, since it does not absorb the prompting information, the improvement is relatively small. We thus infer that the hierarchical prompting is a superior approach for utilizing the hierarchical labels. **3)** Variant 2 ("No coarse labels") barely achieves any improvement over the baseline, though it also increases the same amount of parameters as TransHP. It indicates that the benefit of TransHP is not due to the increase of some trainable tokens. Instead, the coarse class information injected through the prompt tokens matters.

**TransHP gains consistent improvements on more datasets.** Besides the most commonly used dataset ImageNet, we also conduct experiments on some other datasets, *i.e.*, iNaturalist-2018, iNaturalist-2019, CIFAR-100 and DeepFashion. For these datasets, we use two settings, *i.e.*, training from scratch (w/o Pre) and finetuning from the ImageNet-pretrained model (w Pre). The experimental results are shown in Table 1, from which we draw two observations. First, under both settings, TransHP brings consistent improvement over the baselines. Second, when there is no pre-training, the improvement is even larger, especially on small datasets. For example, we note that on the smallest CIFAR-100, the improvement under "w/o Pre" and "w Pre" are $+5.32\%$ and $+1.87\%$, respectively. We infer it is because TransHP considerably alleviates the data-hungry problem of the transformer, which is further validated in Section 4.3.

**TransHP improves various backbones.** Besides the light transformer baseline used in all the other parts of this section, Table 2 evaluates the proposed TransHP on some more backbones, *i.e.*, ViT-B/16 (Dosovitskiy et al., 2021), ViT-L/16 (Dosovitskiy et al., 2021), DeiT-S (Touvron et al., 2021), and DeiT-B (Touvron et al., 2021). We observe that for the ImageNet classification, TransHP gains $2.83\%$, $2.43\%$, $0.73\%$, and $0.55\%$ improvement on these four backbones, respectively.

**Comparison with state-of-the-art hierarchical classification methods.** We compare the proposed TransHP with two most recent hierarchy-based methods, *i.e.* Guided (Landrieu & Garnot, 2021),

Table 2: TransHP brings consistent improvement on various backbones on ImageNet.

| Accuracy (%) | ViT-B/16 | ViT-L/16 | DeiT-S | DeiT-B |
|---|---|---|---|---|
| Baseline | 76.68* | 76.37* | 79.82 | 81.80 |
| TransHP | 79.51 | 78.80 | 80.55 | 82.35 |

∗ The performance of our reproduced ViT-B/16 and ViT-L/16 are slightly worse than 77.91 and 76.53 in its original paper (Dosovitskiy et al., 2021), respectively.

Table 3: Comparison between TransHP and two most recent state-of-the-art methods. We replace their CNN backbones with the relatively strong transformer backbone for fair comparison.

| Accuracy (%) | ImageNet | iNat-2018 | iNat-2019 | CIFAR-100 | DeepFashion |
|---|---|---|---|---|---|
| Baseline | 76.21 | 63.01 | 69.31 | 84.98 | 88.54 |
| Guided | 76.05 | 63.11 | 69.66 | 85.10 | 88.32 |
| HiMulConE | 77.52 | 63.46 | 70.87 | 85.43 | 88.87 |
| TransHP | **78.65** | **64.21** | **71.62** | **86.85** | **89.93** |

HiMulConE (Zhang et al., 2022a). We do not include more competing methods because most prior works are based on the convolutional backbones and are thus not directly comparable with ours. Since the experiments on large-scale datasets is very time-consuming, we only select the most recent state-of-the-art methods and re-implement them on the same transformer backbone (based on their released code). The experimental results are shown in Table 3. It is clearly observed that the proposed TransHP achieves higher improvement and surpasses the two competing methods. For example, on the five datasets, TransHP surpasses the most recent state-of-the-art HiMulConE by $+1.13\%$ (ImageNet), $+0.75\%$ (iNat-2018), $+0.75\%$ (iNat-2019), $+1.42\%$ (CIFAR-100) and $1.06\%$ (DeepFashion), respectively. We also notice that while Guided achieves considerable improvement on the CNN backbones, its improvement over our transformer backbone is trivial. This is reasonable because improvement over higher baseline (*i.e.*, the transformer backbone) is relatively difficult. This observation is consistent with (Zhang et al., 2022a).

### 4.3 TRANSHP IMPROVES DATA EFFICIENCY

We investigate TransHP under the data-scarce scenario. To this end, we randomly select $1/10$, $1/5$, and $1/2$ training data from each class in ImageNet (while keeping the validation set untouched). The results are summarized in Table 4, from which we draw three observations as below:

**F**irst, as the training data decreases, all the methods undergo a significant accuracy drop. This is reasonable because the deep learning method in its nature is data-hungry, and arguably this data-hungry problem is further underlined in transformer (Dosovitskiy et al., 2021). **S**econd, compared with the baseline and two competing hierarchy-based methods, TransHP presents much higher resistance against the data decrease. For example, when the training data is reduced from $100\% \to 10\%$, the accuracy drop of the baseline and two competing methods are $50.97\%$, $50.38\%$ and $46.76\%$, respectively. In contrast, the accuracy drop of the proposed TransHP ($40.72\%$) is significantly smaller. **T**hird, since TransHP undergoes relatively smaller accuracy decrease, its superiority under the low-data regime is even larger. For example, its surpasses the most competing HiMulConE by $1.13\%$, $1.51\%$, $5.21\%$ and $7.17\%$ under the $100\%$, $50\%$, $20\%$ and $10\%$ training data, respectively. Combining all these observations, we conclude that TransHP improves the data efficiency.

### 4.4 TRANSHP IMPROVES MODEL EXPLAINABILITY

We analyze the receptive field of the class token to understand how TransHP reaches its prediction. Basically, the transformer integrates information across the entire image according to the attention map, yielding its receptive field. Therefore, we visualize the attention map of the class token in Fig. 5. For the proposed TransHP, we visualize the attention map at the prompting block (which handles the coarse-class information) and the last block (which handles the fine-class information). For the ViT baseline, we only visualize the attention score map of the the last block. We draw two observations from Fig. 5:

First, TransHP has a different attention pattern compared with the baseline. The baseline attention generally covers the entire foreground, which is consistent with the observation in Dosovitskiy et al.

Table 4: Comparison between TransHP and prior state-of-the-art hierarchical classification methods under the insufficient data scenario. "$N\%$" means using $N\%$ ImageNet training data.

| Accuracy (%) | 100% | 50% | 20% | 10% |
|---|---|---|---|---|
| Baseline | 76.21 | 67.87 | 44.60 | 25.24 |
| Guided | 76.05 | 67.74 | 45.02 | 25.67 |
| HiMulConE | 77.52 | 69.23 | 48.50 | 30.76 |
| **TransHP** | **78.65** | **70.74** | **53.71** | **37.93** |

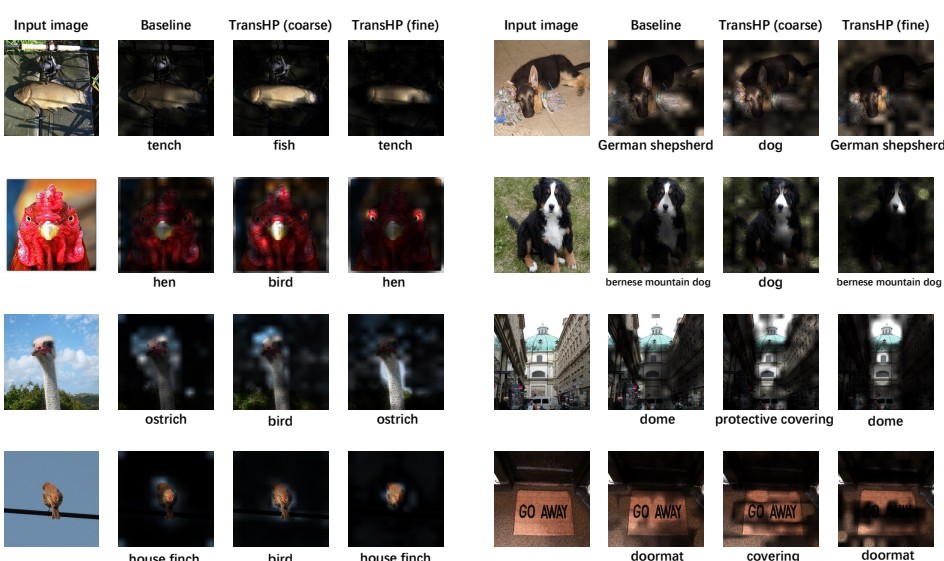

Figure 5: Visualization of the attention map for analyzing the receptive field. For TransHP, we visualize a block before and after the receiving the prompt (*i.e.*, coarse and fine), respectively. The "coarse" block favors an overview for coarse recognition and the "fine" block further filters out the non-relevant regions after receiving the prompt.

(2021). In contrast, in TransHP, although the coarse block attends to the overall foreground as well, the fine block concentrates its attention on relatively small and critical regions, in pace with the "prompting $\rightarrow$ predicting" procedure. For example, given the "hen" image on the second row (left), TransHP attends to the overall foreground before receiving the coarse-class prompt (*i.e.*, the bird) and focuses to the eyes and bill for recognizing the "hen" out from the "bird". Second, TransHP shows better capacity for ignoring the redundant and non-relevant regions. For example, given the "doormat" image on the fourth row (right), TransHP ignores the decoration of "GO AWAY" after receiving the coarse-class prompt of "covering". Similar observation is with the third row (right), where TransHP ignores the walls when recognizing the "dome" out from "protective covering".

## 5 CONCLUSION

This paper proposes a novel Transformer with Hierarchical Prompting (TransHP) for image classification. Before giving its final prediction, TransHP predicts the coarse class with an intermediate layer and correspondingly injects the coarse-class prompt to condition the subsequent inference. An intuitive effect of our hierarchical prompting is: TransHP favors an overview of the object for coarse prediction and then concentrates its attention to some critical local regions after receiving the prompt, which is similar to the human visual recognition. We validate the effectiveness of TransHP through extensive experiments and hope the hierarchical prompting reveals a new insight for understanding the transformers.

ETHIC STATEMENT

Our work strictly follows the ICLR Code of Ethics: *e.g.* (1) The datasets used for evaluating methods are all publicly available and respect others' privacy. (2) Our work mainly focuses on the theoretical contribution, which hardly brings any harm to any people or the natural environment. (3) We try our best to finish the work to uphold high standards of scientific excellence. Under this process, all experimental results are reported accurately and honestly. (4) We acknowledge that there may be bias from the training data, while our proposed method does not include new bias. (5) Our method may fail on out-of-distribution datasets. The proposed method is trained and tested on the same domain, and no guarantee is made about whether the trained models are generalizable.

REPRODUCIBILITY STATEMENT

The proposed TransHP is reproducible. We provide enough method details and training hyperparameters in the main text as well as the appendix. The code will be released based on the acceptance.

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

## A APPENDIX

### A.1 MULTIPLE LAYERS OF HIERARCHY.

We illustrate the TransHP in Fig. 6 when a dataset has multiple layers of hierarchy.

### A.2 COARSE-LEVEL CLASSES OF CIFAR-100

[0]: aquatic mammals, [1]: fish, [2]: flowers, [3]: food containers, [4]: fruit and vegetables, [5]: household electrical devices, [6]: household furniture, [7]: insects, [8]: large carnivores, [9]: large man-made outdoor things, [10]: large natural outdoor scenes, [11]: large omnivores and herbivores, [12]: medium mammals, [13]: non-insect invertebrates, [14]: people, [15]: reptiles, [16]: small mammals, [17]: trees, [18]: vehicles-1, and [19]: vehicles-2.

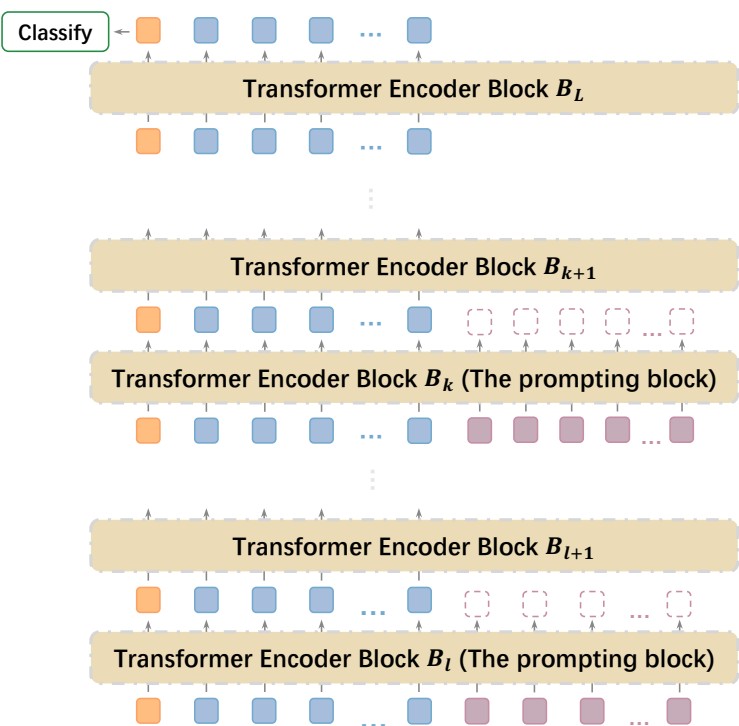

Figure 6: The illustration of TransHP with multiple layers of hierarchy. $k$ and $l$ are two insider layers, and $L$ is the final layer.

Table 5: The balance parameters used for $\mathcal{L}_{coarse}$ of different levels (The last 1 is the balance parameter for the final classification.). "-" denotes that this transformer layer does not have prompt tokens.

| $\lambda$ | 0 | 1 | 2 | 3 | 4 | 5 | 6 | 7 | 8 | 9 | 10 | 11 |
|---|---|---|---|---|---|---|---|---|---|---|---|---|
| ImageNet | 0.1 | 0.1 | 0.1 | 0.1 | 0.1 | 0.15 | 0.15 | 0.15 | 0.15 | 1 | 1 | 1 |
| iNaturalist-2018 | – | – | – | – | – | – | 1 | – | – | – | – | 1 |
| iNaturalist-2019 | – | – | – | – | – | – | 1 | – | – | – | – | 1 |
| CIFAR-100 | – | – | – | – | – | – | – | – | 1 | – | – | 1 |
| DeepFashion | – | – | – | – | – | – | 0.5 | – | 1 | – | – | 1 |

## A.3 IMPLEMENTATION DETAILS

**Datasets**. We evaluate the proposed TransHP on five datasets with hierarchical labels, *i.e.*, ImageNet (Deng et al., 2009), iNaturalist-2018 (Van Horn et al., 2018), iNaturalist-2019 (Van Horn et al., 2018), CIFAR-100 (Krizhevsky et al., 2009), and DeepFashion-inshop (Liu et al., 2016). The hierarchical labels of ImageNet are from WordNet (Miller, 1998), with details illustrated on Mike's website. Both the iNaturalist-2018/2019 have two-level hierarchical annotations: a super-category (14/6 classes) for the genus, and $8,142/1,010$ categories for the species. CIFAR-100 also has two-level hierarchical annotations: the coarse level has 20 classes, and the fine level has 100 classes. DeepFashion-inshop is a retrieval dataset with three-level hierarchy. To modify it for the classification task, we random select $1/2$ images from each class for training, and the remaining $1/2$ images for validation. Both the training and validation set contain 2 coarse classes, 17 middle classes, and $7,982$ fine classes, respectively.

**Training details.** Our TransHP adopts an end-to-end training process. We use a lightweight transformer as our major baseline, which has 6 heads (half of ViT-B) and 12 blocks. The dimension of the embedding and the prompt token is 384 (half of ViT-B). We train it for 300 epochs on 8 Nvidia

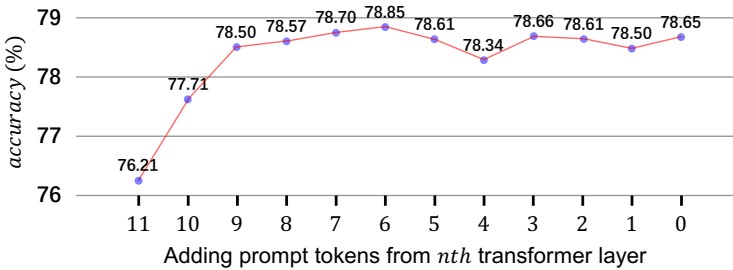

Figure 7: The top-1 accuracy on ImageNet *w.r.t* the transformer layer from which to add prompt tokens. The highest two transformer layers (which do not1 have too coarse-level labels) play an important role.

Table 6: The analysis of the number of coarse-level classes on the CIFAR-100 dataset. "$N$-class" denotes that there are $N$ classes for the coarse-level classification.

| Accuracy (%) | baseline | 2-class | 5-class | 10-class | 20-class |
|---|---|---|---|---|---|
| w/o Pre | 61.77 | 63.34 | 63.12 | 64.47 | 67.09 |
| w Pre | 84.98 | 86.40 | 86.35 | 86.50 | 86.85 |

A100 GPUs and PyTorch (Paszke et al., 2019). The base learning rate is 0.001 with cosine learning rate. We set the batch size, the weight decay and the number of warming up epochs as 1,024, 0.05 and 5, respectively. We have a qualitative principle to set the position for inserting the prompts: if the number of coarse classes is small (large), the position of the corresponding prompting blocks should be close to the bottom (top). Table 5 in Appendix A.4 summarizes the setting of the balance parameters and the position of prompting layers. We note that our setting of the balance parameters is not elaborately tuned, and shows some robustness for TransHP. Moreover, we find that when given many levels of hierarchy (*e.g.*, the ImageNet-1000 has at most 12 levels), the benefit from the highest levels (close to the root) is relatively trivial (See Appendix A.5). Importantly, TransHP only adds small overhead to the baseline. Specifically, compared with the baseline (22.05 million parameters), our TransHP only adds 0.60 million parameters (about $+2.7\%$) for ImageNet. When using ViT-B as the backbone, our TransHP only adds $+1.4\%$ parameters.

### A.4    THE BALANCE PARAMETERS OF DIFFERENT DATASETS.

Please refer to Table 5 for the positions to insert prompt and corresponding balance parameters.

### A.5    IMPORTANCE ANALYSIS OF CLASSIFICATION AT DIFFERENT HIERARCHICAL LEVELS.

From Table 5 (Line 1), each transformer layer is responsible for one level classification. We remove the prompt tokens from the coarsest level to the finest level. In Fig. 7, $n$ denotes that the prompt tokens are added from the $n$th transformer layer. We conclude that only the last two coarse level classifications (arranged at the 9th and 10th transformer layer) contribute most to the final classification accuracy. That means: (1) it is not necessary that the number of hierarchy and transformer layers are equal. (2) it is no need to adjust any parameters from too coarse level hierarchy. (Note that: though the current balance parameter for the 8th transformer layer is 0.15, when it is enlarged to 1, no further improvement is achieved.)

### A.6    ANALYSIS OF THE NUMBER OF COARSE-LEVEL CLASSES

As shown in Appendix A.2, the CIFAR-100 dataset has 20 coarse-level classes. When we combine them into 10 coarse-level classes, we have ([0-1]), ([2-17]), ([3-4]), ([5-6]), ([12-16]), ([8-11]), ([14-15]), ([9-10]), ([7-13]), and ([18-19]). When we combine them into 5 coarse-level classes, we have

Table 7: Comparison between TransHP with the original baseline and the "No prompts" baseline.

| Accuracy (%) | iNat-2018 | iNat-2019 | CIFAR-100 | DeepFashion |
|---|---|---|---|---|
| Baseline (w/o Pre) | 51.07 | 57.33 | 61.77 | 83.42 |
| No prompts (w/o Pre) | 51.88 | 58.45 | 63.78 | 84.23 |
| TransHP (w/o Pre) | **53.22** | **59.24** | **67.09** | **85.72** |
| Baseline (w Pre) | 63.01 | 69.31 | 84.98 | 88.54 |
| No prompts (w Pre) | 63.41 | 70.73 | 85.50 | 89.59 |
| TransHP (w Pre) | **64.21** | **71.62** | **86.85** | **89.93** |

Table 8: The top-1 accuracy of TransHP on some other datasets (besides ImageNet) with standard ViT-B/16 backbone. "w Pre" or "w/o Pre" denotes the models are trained from ImageNet pre-training or from scratch, respectively.

| Accuracy (%) | iNaturalist-2018 | iNaturalist-2019 | CIFAR-100 | DeepFashion |
|---|---|---|---|---|
| ViT-B/16 (w/o Pre) | 52.96 | 58.24 | 62.91 | 84.28 |
| TransHP (w/o Pre) | 54.33 | 60.14 | 69.32 | 86.82 |
| ViT-B/16 (w Pre) | 64.10 | 70.22 | 87.13 | 89.14 |
| TransHP (w Pre) | 66.43 | 73.14 | 88.76 | 90.31 |

([0-1-12-16]), ([2-17-[3-4]), ([5-6-9-10]), ([8-11-18-19]), and ([7-13-14-15]). When we combine them into 2 coarse-level classes, we have ([0-1-7-8-11-12-13-14-15-16]) and ([2-3-4-5-6-9-10-17-18-19]). The experimental results are listed in Table 6.

We observe that: 1) Generally, using more coarse-level classes is better. 2) Using only 2 coarse-level classes still brings over $1\%$ accuracy improvement.

## A.7 THE COMPARISON WITH THE "NO PROMPTS" BASELINE.

In this section, we provide more experiments with the "No prompts" baseline. The detail of the "No prompts" baseline is shown in Fig. 4 (2). The experimental results are shown in Table 7. We find that though "No prompts" baseline surpasses the original baseline, our TransHP still shows significant superiority over this baseline.

## A.8 MORE EXPERIMENTS WITH THE VIT-B/16 BACKBONE.

In this section, we provide more experiments with the standard ViT-B/16 backbone. The experimental results are shown in Table 8. We find that no matter with pre-trained models or without, the TransHP achieves consistent improvement on all these datasets.

## A.9 ADDITIONAL $L_{coarse}$ WITH DEIT.

We introduce the experimental results by only adopting $L_{coarse}$ in DeiT. Note that the $L_{coarse}$ is imposed on the class token as shown in Fig. 4 (2). We find that the TransHP still shows performance improvement compared with only using $L_{coarse}$ on DeiT-S and DeiT-B: compared with DeiT-S ($79.82\%$) and DeiT-B ($81.80\%$), "only with $L_{coarse}$" achieves $79.98\%$ and $81.76\%$ while the TransHP achieves $80.55\%$ and $82.35\%$, respectively.

## A.10 LIMITATION OF TRANSHP.

One limitation of our method is that we presently focus on the image classification task in this paper, while there are some other tasks that are the potential to benefit from hierarchical annotations, e.g.,

semantic segmentation. That being said, we note that our experiments on five datasets are very comprehensive compared with prior hierarchical image classification literature.

## A.11 EFFICIENCY COMPARISON.

Due to the increase of parameters ($+2.7\%$ on our baseline and $+1.4\%$ on ViT-B for ImageNet) and the extra cost of the backward of several $L_{coarses}$, the training time increases by $15\%$ on our baseline and $12\%$ on ViT-B for ImageNet. For inference, the computation overhead is very light. The baseline and TransHP both use around $50$ seconds to finish the ImageNet validation with $8$ A100 GPUs.

