# OpenReview forum: "Hierarchical Prompting Improves Visual Recognition On Accuracy, Data Efficiency and Explainability"
_ICLR.cc/2023/Conference — Submitted to ICLR 2023_

### Official Review · Reviewer_fQu3 · 2022-10-16

**Confidence:** 4
**Correctness:** 3
**Technical Novelty And Significance:** 3
**Empirical Novelty And Significance:** 2
**Recommendation:** 5

**Clarity, Quality, Novelty And Reproducibility:**

The attempt to improve the ViT is incremental innovation. I am concerned about not being able to reproduce the results of the paper. A lot of details in the paper are not clear enough. I have some questions about the implementation details as follows:

1. Did the authors compare whether the training time was increased? What about the inference time?

2. What are the limitations of the method proposed by the authors? How does it need to be modified for the case where there are no hierarchical labels?

Also, I have some questions about the implementation details.

1. I understand the paper. But I am confused about how to train the model proposed in the paper. Do you need to train 𝐩𝐫𝐨𝐦𝐩𝐭𝐢𝐧𝐠 𝐛𝐥𝐨𝐜𝐤 first? Or are 𝐩𝐫𝐨𝐦𝐩𝐭𝐢𝐧𝐠 𝐛𝐥𝐨𝐜𝐤 trained together with the final classification network?

2. After obtaining the Absorbing prompt tokens, what are those tokens input to the final Transformer Block? Is there only one token or multiple tokens as input?


3. "The learning procedure is illustrated in Fig. 2 (a)." in front of Equation 3 is wrong here, right? So, where exactly does it refer to in Figure 2?

4. Are all predefined prompt tokens entered in Equation 3? Will the category of Prompt tokens change?

5. Are Equations (4) and (5) using a simple attention mechanism? The meaning of Equation (6) should not be simply adding softmax and cross-entropy loss. Does the author here just want to discover an Absorbing prompt token?

6. Does the author's interpretability mean the weight of the absorption weight and ratio of the target prompt score?

**Details Of Ethics Concerns:**

I don't have an ethical issue.

**Strength And Weaknesses:**

Advantages:
1. The idea of introducing coarse class labels is straightforward and efficient. It has been verified in previous CNN work.
2. The idea of Prompt fully improved the results of the original baseline in the experiment.
3. The paper's proposed method seems helpful for interpreting the ViT backbone.

Disadvantages:
1. Lack of training details. I am confused about how to train the model proposed in the paper. Is it necessary to train 𝐩𝐫𝐨𝐦𝐩𝐭𝐢𝐧𝐠 𝐛𝐥𝐨𝐜𝐤 first? Or 𝐩𝐫𝐨𝐦𝐩𝐭𝐢𝐧𝐠 𝐛𝐥𝐨𝐜𝐤 and the final classification network are trained together?
2. Lack of analysis of general conditions. If there are no hierarchical labels, can the method proposed in the paper still be applicable? Which improvements are needed?
3. The paper structure and writing are a little confusing. Some of the details are not well explained. For example, the formula symbols are confusing, and the subscripts and subscripts are incorrect.
4. The authors did not discuss the limitations of the method.

**Summary Of The Paper:**

The paper makes the following improvements based on Vision Transformer (ViT).
1. The paper adds a Transformer Block to the existing ViT to predict coarse categories.
2. After obtaining the coarse category results, a simple self-attention mechanism is used to obtain the coarse category weights.
3. Finally, the coarse category is input into the final Transformer Block to get the final result.

The paper has been validated on datasets with hierarchical labels such as ImageNet, iNaturalist-2018, iNaturalist-2019, CIFAR-100, and DeepFashion-inshop.

The paper provides some target prompt weights. The paper argues that this is a manifestation of improving interpretability.

**Summary Of The Review:**

I think there is some innovation in the paper. But there is a lot of ambiguity in organization and presentation. There is also a lack of comparisons of training details and efficiency.

The paper lacks an analysis of the limitations of the proposed approach. Also, there seems to be only some weight for interpretability. A more detailed analysis of interpretability is lacking.

So my opinion is marginally above the acceptance threshold.

---

> ### Author Response · Authors · 2022-11-18
> **Reply to Reviewer 4 (fQu3)**
>
> **Q1: Lack of training details. I am confused about how to train the model proposed in the paper. Is it necessary to train 𝐩𝐫𝐨𝐦𝐩𝐭𝐢𝐧𝐠 𝐛𝐥𝐨𝐜𝐤 first? Or 𝐩𝐫𝐨𝐦𝐩𝐭𝐢𝐧𝐠 𝐛𝐥𝐨𝐜𝐤 and the final classification network are trained together?**
>
> Ans: We apologize for making you feel confused. All the components (backbone, prompts, final classification head) are trained simultaneously in an end-to-end manner. Please first refer to important clarifications (b) for details. We have clarified this in the revision.
>
>
> **Q2: Lack of analysis of general conditions. If there are no hierarchical labels, can the method proposed in the paper still be applicable? Which improvements are needed?**
>
> Ans: Thanks for the question. Please refer to the common concern for the explanation.
>
>
> **Q3: The paper structure and writing are a little confusing. Some of the details are not well explained. For example, the formula symbols are confusing, and the subscripts and subscripts are incorrect.**
>
> Ans: Thanks. We will carefully revise the manuscript.
>
>
> **Q4: The authors did not discuss the limitations of the method.**
>
> Ans: We add our method’s limitations to Appendix A.10.
>
> One limitation of our method is that we presently focus on the image classification task in this paper, while there are some other tasks that are the potential to benefit from hierarchical annotations, e.g., semantic segmentation. That being said, we note that our experiments on five datasets are very comprehensive compared with prior hierarchical image classification literature.
>
>
> **Q5: Did the authors compare whether the training time was increased? What about the inference time?**
>
> Ans: Thanks for the question. Due to the increase of parameters (+2.7% on our baseline and +1.4% on ViT-B for ImageNet) and the extra cost of the backward of several L_coarses, the training time increases by 15% on our baseline and 12% on ViT-B for ImageNet. For inference, the computation overhead is very light. The baseline and TransHP both use around 50 seconds to finish the ImageNet validation with 8 A100 GPUs. We have added this comparison to Appendix A.11.
>
>
> **Q6: After obtaining the Absorbing prompt tokens, what are those tokens input to the final Transformer Block? Is there only one token or multiple tokens as input?**
>
> Ans: Thanks. As shown in Fig. 1 (b), the prompt tokens do not pass to the following transformer block. Therefore, the last transformer block has only the class token and patch token as its input.
>
>
> **Q7: "The learning procedure is illustrated in Fig. 2 (a)." in front of Equation 3 is wrong here, right? So, where exactly does it refer to in Figure 2?**
>
> Ans:  Sorry for this editorial problem. Fig. 2 (a) should be Fig. 2 (i).
>
> **Q8: Are all predefined prompt tokens entered in Equation 3? Will the category of Prompt tokens change?**
>
> Ans: Yes, all the predefined prompt tokens are entered in Eq. 3. Each prompt is responsible for representing a constant category.
>
>
> **Q9: Are Equations (4) and (5) using a simple attention mechanism? The meaning of Equation (6) should not be simply adding softmax and cross-entropy loss. Does the author here just want to discover an Absorbing prompt token?**
>
> Ans: Eq. 4 and Eq. 5 have nothing to do with the attention mechanism. Together with Eq. 3, they make up the training process (as illustrated in Fig. 2(i)). The overall training combines L_coarse and the final fine-grained classification losses.
>
> Eq. 6 formulates the absorption weights (i.e., the attention score) of a specific patch token. From Eq. 6 we further derive the class token’s absorption weights on the prompts (Eq. 7). We investigate the absorption characteristic based on Eq. 7.
>
> **Q10: Does the author's interpretability mean the weight of the absorption weight and ratio of the target prompt score?**
>
> Ans: Thanks for the question. We consider the interpretability from two viewpoints. 1) As you noticed, TransHP automatically assigns the largest absorption weight to the target coarse class (i.e., the on-the-fly predicted coarse class), as illustrated in Fig. 2 (ii) in the manuscript. 2) TransHP favors an overview of the object for coarse prediction and then concentrates its attention on some critical local regions after receiving the prompt, as visualized in Fig. 5. This pattern is similar to the human visual recognition procedure.

---

> ### Author Response · Authors · 2022-12-02
> **Happy to provide additional clarification**
>
> We find you change the score from 6 to 5 without explanation. We would like to hear your feedback and have a further discussion.

---

### Official Review · Reviewer_N53d · 2022-10-24

**Confidence:** 4
**Correctness:** 3
**Technical Novelty And Significance:** 3
**Empirical Novelty And Significance:** Not applicable
**Recommendation:** 6

**Clarity, Quality, Novelty And Reproducibility:**

The clarity is good.
The Quality is fair.
The Novelty is good.
The Reproducibility is good.

**Strength And Weaknesses:**

Strength:
1. The idea of using additive tokens to extract coarse class information to help the fine visual recognition is promising.
2. The visualization of the attention map in Fig.5 is interesting. Still, I wonder whether fine attention dominates the fine-grained classification, e.g., this attention is the key to distinguishing all classes in one coarse category.

Weakness:
1. For prompt tuning, the backbone is frozen while only the prompt token is learnable. However, I find no specific statement about which parameters need to be trained.
I think the proposed TransHP needs to fintune all the models and cannot be termed as 'prompt'.
In Fig.1, the fire symbol used in prompt tuning always means the only tuned parameter while the ice symbol means the frozen parameter.

2. This is not clear whether some additional paramters with coarse labels rather than the proposed hierarchical prompt tokens are the key to the performance gain.
As shown in Table 2, DeiT-S and DeiT-B is only a little inferior compared with the proposed approach. If we adpot the L_coarse in DeiT, whether the performance can be comparable with the proposed TransHP?

3. In "Remark 1" on page 6, the authors use all the tokens to calculate the absorption weight of the target prompt.
Since the absorption weight of the target prompt represents which coarse label the input image is from, then only use the prompt tokens are nature.
I wonder why use all the tokens including the feature tokens to calculate the absorption weight.
Is this a reweighting mechnasim for feature tokens?

4. How the coarse labels are selected and ablation studies on the coarse labels may be necessary, e.g., the number of the coarse labels (flowers and trees can be combined as one coarse label).

5. In this paper, all the prompt tokens are prepend after the feature tokens and trained with soft weighting.
This means the prompt token, which is sigificantly different from the target coarse label, is also used to guide the fine features.
I wonder whether a hard weighting mechaism can serve as a sparse Mixture of experts to help the fine-grained visual recognition and reduce the computation cost.
6. I wonder why a new ViT baseline is created in the main experiment when ViT-small and ViT-base are widely adopted.
Actually, ViT-small has a 384-dimension embedding and 6 heads 12 blocks, which is the same as your proposed new lightweight transformer.

7. From my opinion, the proposed hierarchical prompt tokens can learn the coarse label and align the features.
It seems that this architecture is designed for fine-grained tasks.
However, all the experinments are conducted on datasets with several coarse labels.
For me, the trivial increment on ImageNet (82.80->82.35) at the cost of expensive fintuing is not interesting.
If the proposed hierarchical architecture can achieve great performance gain on fine-grained tasks with tuning only the target prompt token, then this is more meaningful.
So, can the TransHP serves as a good prompt tuning approach for fine-grained tasks?

8. Lcoarse in Eq.5 is confusing, how coarse labels participate in the calculation of loss when only learnable prototypes and prompt tokens are used?
How are the prototypes learned?

**Summary Of The Paper:**

This paper proposes to leverage the injetced prompt token to capture the coarse class feature to benefit the visual recognition.
With the additive corase class labels the proposed approach can outperform the baseline with less than 2% additional parameters.


**Summary Of The Review:**

The idea of using additive tokens to extract coarse class information to help the fine visual recognition is promising.
However, I think finetuning all the parameters is not interesting and some experiments are missing and some statements are confusing.

---

> ### Author Response · Authors · 2022-11-18
> **Reply to Reviewer 3 (N53d)**
>
> We find multiple questions (Q1, Q7, and Q8) are raised due to the same misunderstanding, i.e., TransHP adopts a two-stage pipeline of “pretraining → finetuning”. We apologize for making you feel confused on this point. TransHP is different from the popular prompting technique and actually learns the backbone and prompts simultaneously. We believe after we clarify this detail, you will have no more concerns about Q1, Q7, and Q8.
>
> **Q1: For prompt tuning, the backbone is frozen while only the prompt token is learnable. 1) However, I find no specific statement about which parameters need to be trained. I think the proposed TransHP needs to fine-tune all the models and cannot be termed as 'prompt'. 2) In Fig.1, the fire symbol used in prompt tuning always means the only tuned parameter while the ice symbol means the frozen parameter.**
>
> Ans:
>
> 1) Thank you for reminding us of this difference. Our method does not adopt the usual pipeline of “pre-training the base model → prompt learning’’.  In the proposed TransHP, the base model and the prompt tokens are simultaneously trained from scratch in an end-to-end manner. However, we think TransHP can still be viewed as a prompting technique because different prompts change the mapping function of the transformer and condition the model for different coarse classes. This effect is consistent with the basic connotation of prompting: changing the mapping function of the model by modifying the context of the input. Please refer to the general responses at the beginning for more details.
>
> 2) The fire/ice symbol in Fig. 1 does not mean the tuned/frozen parameter. Instead, we use a fire/ice symbol to indicate that the corresponding prompt is absorbed into the class token with a large/small weight. Specifically, we prepend the prompts for all the coarse classes into the transformer block. During inference, the model predicts the coarse class of the input image and correspondingly decides which prompt should be absorbed (large weight) and which prompts should be ignored (small weights).
>
>
> **Q2: Which is the key to the performance gain, the additional parameter with coarse labels or the proposed hierarchical prompt tokens? In Table 2, DeiT-S and DeiT-B are only a little inferior compared with the proposed approach. If we adopt the L_coarse in DeiT, whether the performance can be comparable with the proposed TransHP?**
>
> Ans: Thanks. We believe the hierarchical information injected through the prompt tokens is the key to our performance gain. In Fig. 4 in the manuscript, we observe that two variants, i.e., adding coarse supervision without prompts and adding additional prompts without coarse supervision respectively achieve +1.37 and +0.21 improvement, respectively, while our method applying coarse supervision on the prompts achieves +2.44 (>1.37+0.21) improvement.  Moreover, visualization in Fig. 5 shows that hierarchical prompting brings a significant (and positive) impact on the receptive field.
>
> In response to your question about applying L_coarse to DeiT, we conduct the corresponding experiment and summarize the results below.  It is observed that on DeiT, using the coarse classification loss (no prompts) barely achieves improvement. In contrast, based on DeiT-B, our TarnsHP increases the accuracy by +0.55%, which is actually not a small improvement for ImageNet classification.
>
> |  Accuracy (%)  | DeiT-S | DeiT-B  |
> |  :----:  | :----: | :----: |
> | baseline | 79.82 | 81.80  |
> | only with L_coarse | 79.98 | 81.76  |
> | TransHP | 80.55 | 82.35 |
>
>
> **Q3: In "Remark 1" on page 6, the authors use all the tokens to calculate the absorption weight of the target prompt. Why not only focus on the relative weights among the prompts? Since the absorption weight of the target prompt represents which coarse label the input image is from, then only using the prompt tokens is natural. I wonder why to use all the tokens including the feature tokens to calculate the absorption weight. Is this a reweighting mechanism for feature tokens?**
>
> Ans: It is because, in Remark 1, we try to figure out the importance of the target prompt to the class token, given that the class token absorbs information from all the patch tokens + prompt tokens. If we only focus on the comparison among all the prompts (i.e., Remark 2), we will not be able to figure this out. Let us assume an extreme case: the absorption weight for the target prompt is much larger than the weight of other prompts, but is much smaller than the weight of the patch tokens. In this case, the target prompt still has little impact on the class token.

---

> > ### Author Response · Authors · 2022-11-18
> > **Reply to Reviewer 3 (N53d) (part2)**
> >
> > **Q4: How the coarse labels are selected and ablation studies on the coarse labels may be necessary, e.g., the number of the coarse labels (flowers and trees can be combined as one coarse label).**
> >
> > Ans: Thanks. We apologize again for missing the prerequisite background on hierarchical image classification. For details, please refer to the response to the common concern at the beginning. In the response to Reviewer 1 Question 1, we conduct the ablation study on the coarse labels by merging the 20 coarse classes into 10, 5, and 2 classes. The experimental results show that using only 2 coarse classes already brings around 1.5% accuracy improvement, and using more coarse-level classes is better.
> > |  Accuracy (%) | w/o Pre | w Pre |
> > |  :----:  | :----:  | :----: |
> > | baseline | 61.77 | 84.98  |
> > | 2-class | 63.34 | 86.40  |
> > | 5-class | 63.12 | 86.35 |
> > | 10-class | 64.47 | 86.50 |
> > | 20-class | 67.09 | 86.85 |
> >
> > We add the experimental results and the analysis to the Appendix A.6 of the revision.
> >
> >
> > **Q5: In this paper, all the prompt tokens are prepended after the feature tokens and trained with soft weighting. This means the prompt token, which is significantly different from the target coarse label, is also used to guide the fine features. I wonder whether a hard weighting mechanism can serve as a sparse Mixture of experts to help fine-grained visual recognition and reduce the computation cost.**
> >
> > Ans: Insightful question. If we replace the soft weighting with a hard (one-hot) weighting for inference, the achieved accuracy slightly decreases but is still higher than the baseline accuracy. For example, on ImageNet, the baseline achieves 76.21% top-1 accuracy, and our TransHP (soft weighting) achieves 78.65% while hard weighting achieves 77.32%. We guess the performance drop is because the inference condition deviates from the training condition.
> >
> >
> > **Q6: I wonder why a new ViT baseline is created in the main experiment when ViT-small and ViT-base are widely adopted. Actually, ViT-small has a 384-dimension embedding and 6 heads 12 blocks, which is the same as your proposed new lightweight transformer.**
> >
> > Ans: Thanks for the question. We note that our ViT baseline actually has exactly the same architecture as the “ViT-small”. However, we did not directly name it as ViT-small because the original paper of ViT [1] does not report “ViT-small”. The popular ViT-small is a third-party re-implementation.
> >
> >
> > **Q7: The proposed hierarchical prompt tokens can learn the coarse label and align the features. It seems that this architecture is designed for fine-grained tasks. However, all the experiments are conducted on datasets with several coarse labels. The trivial increment on ImageNet (81.80->82.35) at the cost of expensive finetuning is not interesting. If the proposed hierarchical architecture can achieve great performance gain on fine-grained tasks by tuning only the target prompt token, then this is more meaningful. So, can the TransHP serve as a good prompt-tuning approach for fine-grained tasks?**
> >
> > Ans: We apologize for making you feel confused. Our method does not require finetuning, as explained in the general response at the beginning. Instead, it is trained from scratch and has almost the same training cost as the baseline. Given that improving the ImageNet classification accuracy is very hard, we think that a +0.55% improvement on a strong baseline (DeiT-B) is valuable. Moreover, when the training data is relatively scarce, the improvement is much larger, e.g., +9.11% on 10% ImageNet.
> >
> >
> > **Q8: L_coarse in Eq.5 is confusing, how do coarse labels participate in the calculation of loss when only learnable prototypes and prompt tokens are used? How are the prototypes learned?**
> >
> > Ans: After we clarify that our method trains the backbone and the prompts simultaneously in an end-to-end manner, we think there will be no more confusion on this point. We apologize again for making you feel confused and will highlight the end-to-end training pipeline in the manuscript.
> >
> >
> > [1] Dosovitskiy, Alexey, et al. "An image is worth 16x16 words: Transformers for image recognition at scale." arXiv preprint arXiv:2010.11929 (2020).

---

> > ### Author Response · Authors · 2022-12-02
> > **Happy to provide additional clarification**
> >
> > We hope our response helps clear up your initial concerns/questions. We would be happy to provide further clarifications where necessary.

---

### Official Review · Reviewer_S4MJ · 2022-10-25

**Confidence:** 5
**Correctness:** 3
**Technical Novelty And Significance:** 2
**Empirical Novelty And Significance:** Not applicable
**Recommendation:** 5

**Clarity, Quality, Novelty And Reproducibility:**

* Novelty

I am concerned about the comparison with one related work VPT.

* Clarity & Reproducibility

See the weaknesses section. The discussion of how to get the values of some hyper-parameters is not clear, which maybe affect the reproducibility.

**Strength And Weaknesses:**

* Strength

1. The proposed method shows consistent improvements on five selected datasets, various backbones, and the data efficiency setting.

2. The authors provide the visualization results of the attention map to demonstrate how the proposed method improves model explainability.

* Weaknesses

First, I recommend the authors discuss the difference (may provide additional experimental results) with a previous related ECCV paper VPT [1]. Both VPT [1] and TransHP inject extra learning prompts into the visual Transformer encoder, which may weaken the contribution of this paper. This paper looks like just add extra intermediate loss supervision to VPT.

Besides, some implementation details of this paper are not clear to me. I have the following questions:

1. How to select the coarse label $y$ in Eq.(5) ? Does it bring additional annotating requirements?

2. What's the default value of the hyper-parameter $M$ (number of the prompts)? How to pick the value of $M$? Does the value of $M$ same for all the datasets?

3. In Eq.(3), the symbol $l$ means the $l$-th transformer block. What's the value of $l$? Does the authors apply the learnable prompts to one block, or all the blocks?

[1] Visual Prompt Tuning, Jia et al, ECCV 2022

**Summary Of The Paper:**

This paper proposes Hierarchical Prompting that follows coarse-to-fine semantic structure. The authors predefine $M$ learnable prompts (coarse classes) and inject them into the Transformer layers. The $M$ learnable prompts are used to predict the coarse-class prototype $S$ and are optimized by a softmax loss $\mathcal{L}_{\text {coarse }}$. The authors also provide the visualization results to show that TransHP gradually focuses on the predicted coarse class during the training phase. Experimental results on five datasets demonstrate the effectiveness of the proposed method.

**Summary Of The Review:**

See the weaknesses section. I hope the authors can address my concerns.

---

> ### Author Response · Authors · 2022-11-18
> **Reply to Reviewer 2 (S4MJ)**
>
> **Q1: Novelty. Please discuss the difference (may provide additional experimental results) with a previous related ECCV paper VPT [1]. Both VPT [1] and TransHP inject extra learning prompts into the visual Transformer encoder, which may weaken the contribution of this paper.**
>
> Ans: The differences between our TransHP and the VPT method are significant and fundamental. To be general, VPT duplicates the success of prompt-based efficient tuning from NLP to computer vision and still belongs to the efficient-tuning paradigm. In contrast, our TransHP has nothing to do with efficient tuning: it exploits the semantic hierarchy into prompts and improves image classification under the train-from-scratch paradigm. To be more concrete, our TransHP is significantly different from VPT (and previous efficient-tuning methods based on prompting) regarding four aspects, i.e., the objective, the training pipeline, the inference method, and the semantic meaning of the prompts. We have added VPT to the related works in the revised manuscript. We explain these differences below:
>
> **Objective:** VPT explores an efficient-tuning method for computer vision, inspired by similar success in NLP. These methods try to improve the efficiency of adapting an already-trained model to novel downstream tasks. In contrast, TransHP basically learns all the parameters from scratch (although we may also use a pre-trained model for initialization, as well).  We aim to improve the recognition accuracy and data efficiency (under the train-from-scratch paradigm).
>
> **Training pipeline:** VPT (and other prompt-based efficient-tuning methods) adopt the two-stage pipeline. They first pre-train a (large) backbone model on pretext tasks. Then they fix the backbone and only learn the prompts to adapt the model to novel downstream tasks. In contrast, in TransHP, the backbone and the prompts are learned simultaneously.
>
> **Inference method:** After learning multiple sets of prompts for different tasks, VPT (and other prompt-based efficient-tuning methods) manually choose a prompt set for a corresponding downstream task. In contrast, TransHP prepends all the prompts into the model and makes the model itself decide which prompt is useful on-the-fly.
>
> **The semantic meaning of the prompts:** The VPT authors claim that they “tried to find the nearest image patches with the learned prompt embeddings, but they couldn't find any semantic meaningful results.” (https://github.com/KMnP/vpt/issues/16). In contrast, each prompt in TransHP well represents a corresponding coarse class (Fig. 1 (b) and illustrated in Fig. 2 (ii)) and thus has a definite semantic meaning.
>
>
> **Q2: How to select the coarse label $y$ in Eq.(5) ? Does it bring additional annotating requirements?**
>
> Ans: We directly use the coarse labels provided by the datasets. All the employed datasets provide hierarchical annotations. Moreover, when constructing the dataset, adding additional coarse labels barely increases any manual burden. For more details, please refer to the general response to the common concern at the beginning. As for the coarse-label details of the employed datasets in this paper, please refer to the “4.1 Implementation Details Datasets”.
>
>
> **Q3: What's the default value of the hyper-parameter $M$ (number of prompts)? How to pick the value of $M$? Does the value of $M$ the same for all the datasets?**
>
> Ans: $M$ equals the number of coarse classes, i.e., one prompt for each coarse class. For example, for CIFAR-100, $M$ equals $20$.
>
>
> **Q4: In Eq.(3), the symbol $l$ means the $l$-th transformer block. What's the value of $l$? Do the authors apply the learnable prompts to one block or all the blocks?**
>
> Ans: Thanks for these two questions.
>
> (1) The first concern is about the position of the prompting block and is answered in the “general responses” part at the beginning. Moreover, we summarize the empirically-optimized values of $l$ in Table 5 in Appendix A.4. For example, $l=8$ for CIFAR-100. We also note that the achieved accuracy is robust to $l$ to some extent, e.g., on CIFAR-100, setting $l$ to 7 and 8 achieves close results.
>
> (2) It depends on the hierarchy level of the datasets because each coarse level occupies a single block. If the dataset has only two levels of hierarchy (i.e., 1 level for the coarse classes), we apply the learnable prompts to a single block (e.g., we only use the 8-th block as the prompting block for CIFAR-100). The ImageNet dataset has a 12-level hierarchy and thus requires employing all 11 blocks before the last one.

---

> ### Author Response · Authors · 2022-12-02
> **Happy to provide additional clarification**
>
> We hope our response helps clear up your initial concerns/questions. We would be happy to provide further clarifications where necessary.

---

### Official Review · Reviewer_RDBe · 2022-10-25

**Confidence:** 4
**Correctness:** 3
**Technical Novelty And Significance:** 3
**Empirical Novelty And Significance:** 2
**Recommendation:** 5

**Clarity, Quality, Novelty And Reproducibility:**

There are some minor details missing in the paper but the paper is overall easy to follow. The proposed method is interesting and reproducible.

**Strength And Weaknesses:**

Strength
1. The idea of using prompts to learn hierarchical information is interesting.
2. The method is evaluated on multiple dataset and backbones and shows better performance than baselines.
3. The paper is easy to follow.

Weakness
1. The paper didn't explain how to determine the coarse classes. It seems that the datasets used in the paper all have defined hierachical labels, which could be directly used for the proposed method. What if there are no hierarchical labels? Do we have to manually label it? Also, what is the influence of the coarse labels on the final performance. For example, if the dataset can only be divided into very few coarse labels, how much improvement the method could get?
2. The paper didn't explain which layer to add the prompt and what is the effect of different layers.
3. The author mentioned multiple layers of hierarchy, it is better to explain this in the method and Fig. 1.
4. In Fig. 4, the no prompt variant can also improve over the baseline. (1) The author should explain more details about the implementation of this variant. (2) I think this variant should be used as the baseline which the proposed method should compare with in all the experiments. The naive baseline don't use hierarchical labels thus it is not fair.
5. In Table 1, some results seem not very strong. E.g., with ImageNet pre-training, the baseline ViT can only achieve 84.98% top-1 accuracy. The public available results of ViT-B with Imagenet pretraining on CIFAR-100 is around 94%. I understand the author used a smaller ViT here. Thus I suggest the author to use ViT-B as the baseline which makes the results stronger and more convincing.
6. In Fig.5, it is better to provide the coarse and fine label of each image.
7. The method seems to be limited to ViT backbone and image classification.

**Summary Of The Paper:**

This paper proposed a method to learn hierarchical information for image classification. The proposed method introduces prompt tokens into the intermediate layers in ViT, and these prompt tokens are used to make a prediction about the coarse class of the input image. This allows the prompts to learn coarse class information and transfer this information with other tokens in the self-attention layers. The method is evaluated on multiple datasets and backbones and shows better performance than baselines.

**Summary Of The Review:**

I think the proposed idea of using prompts to learn hierarchical information is interesting. But I am not how generalized is the proposed method. I also think the experimental evaluation could be improved.

---

> ### Author Response · Authors · 2022-11-18
> **Reply to Reviewer 1 (RDBe)**
>
> **Q1: (a) The paper didn't explain how to determine the coarse classes. It seems that the datasets used in the paper all have defined hierarchical labels, which could be directly used for the proposed method. What if there are no hierarchical labels? Do we have to manually label it? (b) Also, what is the influence of the coarse labels on the final performance? For example, if the dataset can only be divided into very few coarse labels, how much improvement could the method get?**
>
> Ans:  Thanks for the question. For the questions in (a), please refer to our answer to the common concern. As for (b), we think it is an interesting question and deserves investigation. In response,  we reduce the coarse classes of CIFAR-100 through class-merging and get three different settings, i.e., 10 coarse classes, 5 coarse classes, and only 2 coarse classes (please refer to Appendix A.6 for the merging details). The corresponding results are summarized below:
>
> |  Accuracy (%) | w/o Pre | w Pre |
> |  :----:  | :----:  | :----: |
> | baseline | 61.77 | 84.98  |
> | 2-class | 63.34 | 86.40  |
> | 5-class | 63.12 | 86.35 |
> | 10-class | 64.47 | 86.50 |
> | 20-class | 67.09 | 86.85 |
>
> We observe that: 1) Generally, using more coarse-level classes is better. 2) Using only 2 coarse-level classes still brings over 1% accuracy improvement.
>
>
> **Q2: The paper didn't explain which layer to add the prompt and what is the effect of different layers.**
>
> Ans: We apologize for missing this important implementation detail. Currently, we only have a qualitative principle: if the number of coarse classes is small (large), the position of the corresponding prompting blocks should be close to the bottom (top). Given this principle, we still need empirical optimization to choose the position. Please refer to the important clarifications at the beginning for the detailed setting. We have added this clarification to “Appendix A.3 Training details”, and the detailed setting for all datasets is in Table 5 of Appendix A.4.
>
>
> **Q3: The author mentioned multiple layers of hierarchy, it is better to explain this in the method and Fig. 1.**
>
> Ans: Thanks for the advice. We add “*Note that TransHP may have several prompting blocks, and we only add one in (b) for demonstration.*” to Fig. 1, and “*Note that TransHP may have several prompting blocks. We only introduce one prompting block in this section, and other blocks follow a similar procedure. The whole picture is demonstrated in Appendix A.1*” to the method. Also, we add a new figure to demonstrate multiple layers of hierarchy in Appendix A.1.
>
>
> **Q4: In Fig. 4, the no prompt variant can also improve over the baseline. (1) The author should explain more details about the implementation of this variant. (2) I think this variant should be used as the baseline which the proposed method should compare with in all the experiments. The naive baseline doesn't use hierarchical labels thus it is not fair.**
>
> Ans: Thanks for the questions. We agree that the “no prompts’’ variant is important and have added more results for comparison (on iNaturalist, CIFAR-100, DeepFashion). However, we respectfully disagree with the point of using it as the baseline. There are two reasons: 1) It is a standard experimental setup in hierarchical image classification to use the model trained with only fine labels as the baselines. All competing methods [1, 2] report their improvement over the naive baseline with no hierarchical labels. 2) This variant should not be taken for granted. To the best of our knowledge, hierarchical image classification based on the Transformer (using coarse labels to supervise the hidden states of the class token, in particular) has never been explored.

---

> > ### Author Response · Authors · 2022-11-18
> > **Reply to Reviewer 1 (RDBe) (part2)**
> >
> > **Q5: In Table 1, some results seem not very strong. E.g., with ImageNet pre-training, the baseline ViT can only achieve 84.98% top-1 accuracy (on CIFAR-100). The publicly available results of ViT-B with Imagenet pretraining on CIFAR-100 is around 94%. I understand the author used a smaller ViT here. Thus I suggest the author use ViT-B as the baseline which makes the results stronger and more convincing.**
> >
> > Ans: We note that our implementation with 84.98% top-1 accuracy on CIFAR-100 is reasonable and is actually comparable with ViT-B, while the 94%-accuracy model you mentioned is due to extraordinarily large model size and large-scale pretraining. Specifically, in ViT [3], the reported results on CIFAR-100 are 87.13% (ViT-B/16) and 86.31% (ViT-B/32). Our baseline with 84.98% top-1 accuracy is slightly lower because of a smaller backbone (half channels) and smaller input image (224 vs. 384). In contrast, the 94% accuracy is achieved with much larger backbones (ViT-H/14 and ViT-L/16) and through much larger pre-training (JFT and ImageNet21K), as well as larger input (384).
> >
> > During rebuttal, we add comparisons based on the standard ViT-B backbone. It is observed that the TransHP achieves consistent improvement on all these datasets.
> > |  Accuracy  (%) | iNaturalist-2018 | iNaturalist-2019  | CIFAR-100 | DeepFashion  |
> > |  :----:  | :----:  | :----: | :----: | :----: |
> > | ViT-B/16 (w/o Pre) | 52.96 | 58.24 | 62.91 | 84.28 |
> > | TransHP (w/o Pre) | 54.33 | 60.14 | 69.32 | 86.82  |
> > | ViT-B/16 (w Pre) | 64.10 | 70.22 |87.13 | 89.14 |
> > | TransHP (w Pre) | 66.43 | 73.14 | 88.76 | 90.31  |
> >
> >
> > **Q6: In Fig.5, it is better to provide the coarse and fine labels of each image.**
> >
> > Ans: Thanks for this good advice. We have added the coarse and fine labels of each image to Fig. 5.
> >
> >
> > **Q7: The method seems to be limited to the ViT backbone and image classification.**
> >
> > Ans: We indeed base our experiments on multiple editions of the ViT/DeiT backbones and focus on image classification. We note that many network exploration methods [4,5] use ViT (the most popular transformer for computer vision) as the backbone. Moreover, since TransHP is based on the general attention mechanism rather than any specific design in ViT, we think it has the potential to cooperate with other backbones.
> > We also note that image classification is one of the most fundamental computer vision tasks. Compared with prior hierarchical image classification literature [1,2], our experiments on 5 datasets (4 large-scale datasets + CIFAR-100) are comprehensive.
> >
> >
> >
> >
> > [1] Loic Landrieu and Vivien Sainte Fare Garnot. Leveraging class hierarchies with metric-guided prototype learning. In British Machine Vision Conference (BMVC), 2021.
> >
> > [2] Shu Zhang, Ran Xu, Caiming Xiong, and Chetan Ramaiah. Use all the labels: A hierarchical multi-label contrastive learning framework. In Proceedings of the IEEE/CVF Conference on Computer Vision and Pattern Recognition, pp. 16660–16669, 2022a.
> >
> > [3] Dosovitskiy, Alexey, et al. "An image is worth 16x16 words: Transformers for image recognition at scale." arXiv preprint arXiv:2010.11929 (2020).
> >
> > [4] Wang W, Xie E, Li X, et al. Pyramid vision transformer: A versatile backbone for dense prediction without convolutions[C]//Proceedings of the IEEE/CVF International Conference on Computer Vision. 2021: 568-578.
> >
> > [5] Mehta S, Rastegari M. MobileViT: Light-weight, General-purpose, and Mobile-friendly Vision Transformer[C]//International Conference on Learning Representations. 2021.

---

> ### Author Response · Authors · 2022-12-02
> **Happy to provide additional clarification**
>
> We hope our response helps clear up your initial concerns/questions. We would be happy to provide further clarifications where necessary.

---

### Author Response · Authors · 2022-11-18
**General responses: Summarization of the updates**

Besides the response to the above common concern, there are some other major/important updates to our manuscript, as summarized below:

**(i) Additional experiments:**

(a) Analysis of the number of coarse classes on the CIFAR-100 dataset in Appendix A.6 (@ Reviewer 1 (RDBe) and 3 (N53d)). It shows that when we reduce the number of coarse classes through class-merging, the fine-grained recognition accuracy decreases.

(b) The comparison with the “No prompts” baseline in Appendix A.7 (Fig. 4(2), which has coarse supervision as well) on more datasets (@ Reviewer 1 (RDBe)). It shows that TransHP consistently surpasses this baseline by clear margins on all five datasets.

(c) Adding experiments with the ViT-B/16 backbone on iNaturalist-2018, iNaturalist-2019, CIFAR-100, and DeepFashion (@ Reviewer 1 (RDBe)) in Appendix A.8. It further validates the effectiveness of TransHP.

(d) The experimental results of only applying additional L_coarse to DeiT-S and DeiT-B backbone in Appendix A.9 (@ Reviewer 3 (N53d)). It shows that with stronger baselines, our TransHP still shows significant superiority.

**(ii) Important clarifications:**

(a) *The position for inserting the prompts, i.e., the position of the prompting block (@ Reviewer 1 (RDBe) and 2 (S4MJ)).* We have to admit that we are still unable to set up a quantitative rule for deciding the position. Currently, we only have a qualitative principle: if the number of coarse classes is small (large), the position of the corresponding prompting blocks should be close to the bottom (top). Given this principle, we still need empirical optimization to choose the position. For example, for iNaturalist-2019 (6 coarse classes) / CIFAR-100 (20 coarse classes), we respectively choose the 6-th / 8-th transformer block for inserting the prompts (after empirical optimization). As for the ImageNet dataset which has 12 level annotations from coarse to fine, we naturally arrange these levels into the 1st to the 12-th block in ascending order. We have added this detail to the “training details” (Appendix A. 3).

(b) *Our method does not adopt the usual pipeline of “pre-training the base model → prompt learning’’ but can still be viewed as a prompting technique. In the proposed TransHP, the base model and the prompt tokens are simultaneously trained from scratch in an end-to-end manner.* There is no more fine-tuning after this training. In the manuscript, we might have not highlighted this point clearly enough and thus caused confusion for Reviewer 3 (N53d), considering our method incurs “expensive finetuning” and “should not be termed as prompt”. Reviewer 4 also has a similar confusion on this point.

Specifically, the prompting technique usually consists of two stages, i.e., pre-training a base model and then learning the prompts for novel downstream tasks. When learning the prompt, the pre-trained model is usually frozen, as reminded by Reviewer 3 (N53d). This pipeline is different from our end-to-end pipeline. In spite of this difference, we think TransHP can still be viewed as using prompts because our prompts condition a shared transformer for different coarse classes. This is consistent with the keynote of prompting: the prompts change the mapping function of the deep model by modifying the context of the model input and do not change the model parameters. We have added the above comparison and discussion to the related works.

**Given that all the authors positively recognize our contribution, we hope the reviewers will find our paper acceptable based on the above clarifications and the point-to-point responses below.**

---

### Author Response · Authors · 2022-11-18
**General responses: A common concern on the coarse label**

We thank all the reviewers for their valuable comments. We notice there is a common question regarding how to obtain the coarse labels. We take this question seriously and apologize for missing an important prerequisite background, i.e., the hierarchical image classification task. In hierarchical image classification, it is a standard setup to provide hierarchical annotations (coarse labels + fine labels). This setup is accommodated by many public image classification datasets (e.g., ImageNet, iNaturalist, etc.). We have carefully revised the introduction part by adding a new (the 3rd) paragraph to the manuscript:

*“Specifically, exploiting the underlying semantic hierarchy to improve visual recognition has attracted great research interest and yielded several popular tasks, e.g., hierarchical image classification and hierarchical semantic segmentation. Considering that classification is fundamental for many computer vision tasks, this paper focuses on hierarchical image classification. Many popular image classification datasets (e.g., ImageNet and iNaturalist) can well accommodate this task because they already provide hierarchical annotations (``coarse + fine’’ labels). Compared with prior literature on this topic, our method has significant differences due to the employed prompting mechanism. Please refer to Section 2 (Related Works) for a detailed comparison.”*

We also note that in realistic applications, preparing a new dataset with hierarchical labels is no more time-consuming than annotating only the fine labels. It is because the coarse labels do not need manual annotation. Instead, given the fine labels, we can automatically assign coarse labels using a pre-defined hierarchy. For example, the hierarchical labels of ImageNet are generated based on WordNet [1] (with an intuitive visualization on the website [2]). Therefore, obtaining the coarse labels is convenient.

[1] George A Miller. WordNet: An electronic lexical database. MIT Press, 1998.

[2] https://observablehq.com/@mbostock/imagenet-hierarchy

---

### Author Response · Authors · 2022-12-04
**Further highlight our key contribution and novelty**

The rolling rebuttal period is ending soon. We would like to further highlight our key contribution and novelty, and hope the reviewers will find our paper worthy to be shared with the community. We note an important background, i.e., using semantic hierarchy to improve visual recognition has already been recognized as valuable and yields many popular topics, e.g., hierarchical image classification and hierarchical semantic segmentation. Under this background, this paper focuses on hierarchical image classification and makes the following contribution:

1) Compared with prior hierarchical image classification literature, our method is the first to inject the hierarchical information as prompt / hint. Injecting hierarchical information as prompt is actually similar to the human visual recognition process. Specifically, given an input image, our TransHP predicts its coarse class in an intermediate block, and correspondingly injects the target prompt. The injected target prompt modifies the sub-sequential feature extraction, so that the model can focus on the details for discriminating the descendant sibling classes. Consequently, TransHP substantially improves classification accuracy and outperforms prior state-of-the-art hierarchical image classification methods.

2) Compared with prior prompting techniques, our TransHP is novel as well, regarding the objective, training and inference. Objective: prior prompting methods usually use prompting to adapt a pretrained model to novel down-stream tasks. In contrast, our TransHP basically learns all the parameters from scratch and aims to improve the recognition accuracy and data efficiency. There is no pretraining or task-adaptation. Training and inference: prior prompting methods usually adopt the “pretraining -> learning prompt” pipeline, while our TransHP learns the backbone and prompts simultaneously. During inference, prior methods usually select a single prompt (or partial prompts) for a corresponding task, while our TransHP prepends all the prompts and makes the model itself to decide which prompt is important.

We note that our submission received mixed ratings (5566) at the beginning, and an important reason for those negative rating is that we forgot to introduce the background: how do derive the coarse labels.  During rebuttal, we clarified this confusion: many image classification datasets (e.g., imagenet, iNaturalist) are indeed created with hierarchical labels, and generating hierarchical labels barely incurs manual cost. Given that most reviewers recognized our method as being novel and interesting, we sincerely hope you will raise your scores and recommend sharing our paper to the community in ICLR 2023.

Best wishes,

Authors.

---

### Decision · Program_Chairs · 2023-01-20

**Decision:**

Reject

**Justification For Why Not Higher Score:**

The manuscript in its current form fails to demonstrate the advantage of hierarchical prompting over other alternatives through convincing comparisons and discussions.


**Justification For Why Not Lower Score:**

N/A

**Metareview: Summary, Strengths And Weaknesses:**

This paper was reviewed by four experts in the field. The reviewers raised many concerns regarding the paper: 1) Several important details regarding architecture selection, training paradigm, and hyper-parameter search are missing and unclear. 2) The additional annotation and training costs incurred by the requirement of hierarchical labels should be discussed. Considering the reviewers' concerns, we regret that the paper cannot be recommended for acceptance at this time. The authors are encouraged to consider the reviewers' comments when revising the paper for submission elsewhere.

**Summary Of Ac-Reviewer Meeting:**

N/A